# The Influence of Age on Cerebral Tissue Oxygenation in Vasovagal Syncope and Orthostatic Hypotension

**DOI:** 10.3390/jcm11154302

**Published:** 2022-07-25

**Authors:** Isabella Kharraziha, Parisa Torabi, Madeleine Johansson, Richard Sutton, Artur Fedorowski, Viktor Hamrefors

**Affiliations:** 1Department of Clinical Sciences, Lund University, 202 13 Malmö, Sweden; parisa.torabi@gmail.com (P.T.); madeleine.johansson@med.lu.se (M.J.); r.sutton@imperial.ac.uk (R.S.); artur.fedorowski@med.lu.se (A.F.); viktor.hamrefors@med.lu.se (V.H.); 2Department of Clinical Physiology, Skåne University Hospital, 214 28 Malmo, Sweden; 3Department of Cardiology, Skåne University Hospital, 214 28 Malmo, Sweden; 4National Heart and Lung Institute, Imperial College, Hammersmith Hospital Campus, London W12 0HS, UK; 5Department of Cardiology, Karolinska University Hospital, 171 64 Solna-Stockholm, Sweden

**Keywords:** vasovagal syncope, orthostatic hypotension, cerebral oximetry, head-up tilt, NIRS, aging

## Abstract

Age-related physiological impairment increases susceptibility to syncope. We tested the hypotheses that cerebral oxygenation during orthostatic provocation, as well as the level at which syncope occurs, differs according to age. Non-invasive hemodynamic monitoring and cerebral oximetry were applied during a head-up tilt test in 139 patients with vasovagal syncope (mean (SD) 45, (17) years, 60%-female); 121 patients with orthostatic hypotension (61.4 (19.2) years, 49.6%-female); and 82 patients with a negative head-up tilt test (45 (18) years, 61%-female). Group differences in cerebral tissue oxygenation levels and systolic blood pressure were assessed in supine at 3 and 10 min of orthostatic provocation, 30 s before (i.e., presyncopal phase) and during syncope in age groups of <30, 30–60, and >60 years. During the head-up tilt test, cerebral tissue oxygenation at the presyncopal phase decreased with age, both in patients with vasovagal syncope (<30 years: 66.9 ± 6.2, 30–60: 64.5 ± 6.1, >60: 62.2 ± 5.8%; *p* = 0.009) and orthostatic hypotension (<30: 67.4 ± 4.4, 30–60: 61.6 ± 6.2, >60: 57.5 ± 3.9; *p* < 0.001). Mean systolic blood pressure at the presyncopal phase did not differ according to age. Cerebral oxygenation prior to syncope in older individuals with vasovagal syncope and orthostatic hypotension is lower compared with younger individuals independently of systolic blood pressure. This suggests that the level of cerebral oxygenation at which syncope is elected is lower in older individuals.

## 1. Introduction

Aging is associated with changes in the cardiovascular system, such as impairment of heart rate (HR), blood pressure (BP) and cerebral blood flow (CBF) control [1,2]. These changes in combination with multi-morbidity and polypharmacy may contribute to increased susceptibility to syncope and orthostatic intolerance in the elderly [3]. Syncope and orthostatic intolerance comprise a variety of clinical syndromes ranging from benign sporadic episodes such as vasovagal syncope (VVS), to severely disabling symptoms of autonomic dysfunction such as pronounced orthostatic hypotension (OH) [3]. Syncope may reduce quality of life and accounts for a substantial number of emergency admissions in both younger and older persons. In the older person, syncope carries a high morbidity and mortality [3].

Cerebral autoregulation (CA), which maintains a constant cerebral perfusion pressure over a wide range of systemic blood pressures, has been shown to be impaired during presyncope, and this impairment probably contributes to a sudden fall in CBF, ultimately resulting in loss of consciousness [4]. According to previous research, first syncope incidence has a bimodal distribution, with the highest incidence among subjects below 30 and above 60 years of age, respectively [5]. It has been hypothesized that an age-related impairment of CA may account for these differences, especially the increased incidence of syncope after 60 years [6]. Cerebral oximetry non-invasively measures cerebral tissue oxygenation (SctO2) and has been shown to be informative in syncope evaluation on a head-up-tilt-test (HUT) [7]. Baseline SctO2 decreases with aging [8] but it is unknown whether the decrease in SctO2 is related to increased susceptibility to syncope during orthostatic provocation. By measuring SctO2 during HUT, age-related differences in SctO2 can be studied, together with their impact on developing reflex syncope and loss of consciousness. We tested the hypotheses that (A.) SctO2 during orthostatic provocation is influenced by age and (B.) syncope occurs at a different SctO2 level according to age.

## 2. Materials and Methods

### 2.1. Study Population

This study was based on the SYSTEMA cohort, which has been described previously [9]. Patients that were evaluated for syncope and orthostatic intolerance (n = 2557) at Skåne University Hospital, Malmö, Sweden during 2008–2020 were included. In short, all patients underwent cardiovascular autonomic testing, including HUT. From 2013, cerebral oximetry has been routinely performed during HUT. Patients with VVS, OH and normal response to HUT were selected, based on the clinical interpretation of the HUT response, after excluding those with missing values or inconsistent cerebral oximetry signal. There were 139 patients with VVS: 121 with OH (39 classical; 82 delayed OH); and 82 patients with a normal response to HUT (negative) after nitroglycerine administration who were included as a reference group. Patients with an abnormal response to carotid sinus massage were excluded from this study. Data on comorbidities were retrieved from the patient’s medical record at the time of HUT. A flow-chart illustrating the selection of study subjects is shown in Figure 1. All patients gave written informed consent. The study was approved by the regional ethical review board in Lund (DNR 08/82; 20 May 2008) and all procedures were performed in accordance with the Helsinki Declaration.

### 2.2. Head-Up Tilt Test

Head-up tilt was performed according to the Italian protocol, including sublingual nitroglycerine (400 mcg) administration after 20 min, if syncope or presyncope had not yet occurred [10]. Arterial BP was continuously recorded using a photoplethysmographic device (Nexfin, BMEYE, Amsterdam, The Netherlands; or Finapres Nova, Finapres Medical Systems, PH Enschede, The Netherlands), with a time-synchronized electrocardiogram. 

### 2.3. Cerebral Oximetry

Regional and relatively superficial cerebral tissue oxygenation was measured with near-infrared spectroscopy (NIRS) as previously described [11]. NIRS is a non-invasive method measuring mixed blood oxygen saturation levels in cerebral tissue by determining the ratio of oxygenated hemoglobin to total hemoglobin. It shows a proportional mix of arterial and venous blood in the outer regions of the frontal hemispheres [7]. The normal range of SctO2 is 60–80% [12]. SctO2 was assessed using the Fore-Sight absolute cerebral oximeter (CAS Medical Systems Inc., Branford, CT, USA), which was used in previous SYSTEMA cohort studies [11,13]. Absolute cerebral oximetry and hemodynamic parameters were measured simultaneously and were time-synchronized. 

### 2.4. Definition of Diagnosis Groups

VVS was defined as a reproduction of syncope that was associated with a characteristic pattern of pronounced hypotension, bradycardia, or asystole. OH was defined as a sustained decrease in systolic blood pressure (SBP) ≥ 20 mmHg and/or decrease in diastolic BP ≥ 10 mmHg, or SBP < 90 mmHg [3].

### 2.5. Definition of Variables and Statistical Analyses

The SctO2 values were collected in the supine position, and at three and ten minutes of HUT. The presyncopal phase of SctO2, SBP and HR were calculated as the mean value of 15–45 s prior to syncope. “SctO2 syncope” was defined as the lowest measured value during syncope. Minimum SctO2 was defined as the lowest value at any time, before reflex activation in HUT. An example of a patient during HUT and included variables is shown in Figure 2. SctO2 delta presyncope, syncope and minimum were calculated as the difference between SctO2 when supine and presyncopal phase, syncope and minimum, respectively. 

Group differences between VVS, OH and subjects with negative HUT were compared using a one-way ANOVA for continuous data and Pearson’s chi2-test for dichotomous data. Univariable linear regression models were applied to study the associations between SctO2 or SBP (dependent variable) and age (independent variable). Multivariable-adjusted linear regression models were also constructed, adjusting for sex and concurrent SBP or SctO2, as appropriate, at time of measurement. When SctO2 was chosen as the dependent variable, the analyses were adjusted for sex and concurrent SBP, whereas if SBP was chosen as the dependent variable, the analyses were adjusted for concurrent SctO2. The proportion of smokers within different age groups was analyzed with Pearson’s chi2-test. An independent samples t-test was performed to study differences in supine SctO2 among smokers vs. non-smokers. 

Patients with VVS, OH and negative HUT were divided into three age groups i.e., <30 years, 30–60 years and >60 years, according to previous research suggesting that first syncope incidence has a bimodal distribution, with a high incidence among subjects below 30 years and above 60 years of age [5]. Differences in mean SctO2 during HUT according to different age groups (<30 years, 30–60 years and >60 years) were assessed using a one-way ANOVA after testing for the homogeneity of variance. If the homogeneity of variance had a significance level of <0.05, a Welch test was conducted instead. Either Tukey’s multiple comparisons test or Games–Howell comparisons post-hoc test were performed to study the differences between the three age groups. The SctO2 values were assessed as normally distributed within VVS, OH and negative HUT patients after visual inspection of the distribution plots. The data were analyzed using SPSS software version 25 (SPSS, Chicago, IL, USA). A *p*-value of <0.05 was considered significant for all tests.

## 3. Results

### 3.1. Study Population Characteristics

The baseline characteristics are shown in Table 1. The mean age of patients with OH was higher compared to VVS patients and patients with negative HUT. Further details regarding prescribed medications can be found in Appendix A. Among the VVS patients, 26 patients had spontaneous VVS and the remaining 113 had nitroglycerine-induced VVS. Results on smoking habits according to different age groups are found in Appendix A. There was no difference in supine SctO2 levels among smokers vs. non-smokers (Appendix A). 

### 3.2. Association of SctO2 during HUT with Age and SBP

In VVS patients, age was inversely associated with SctO2 during the presyncopal phase (β = −0.096 per year; *p* = 0.001) after adjusting for concurrent SBP and sex. However, the decrease in SctO2 from supine to presyncopal phase (delta presyncope) was not associated with age after adjusting for sex and concurrent SBP (β = 0.026 per year, *p* = 0.315). In addition, SBP in VVS patients was associated with age during all time points, except for SBP during the presyncopal phase (Appendix A).

Among OH patients, increasing age was associated with lower SctO2 in supine position, after 3 min and 10 min of tilt, in presyncopal phase and minimum SctO2. However, after adjusting for sex and concurrent SBP, no significant association was seen between age and delta SctO2 presyncope (β = 0.070 per year, *p* = 0.052). Furthermore, the SBP levels in OH patients were associated with age after adjusting for sex and concurrent SctO2 during all time points, except for SBP during the presyncopal phase (Appendix A). Among patients with negative HUT there was an association between SctO2 in the supine position and age (β = −0.085 per year; *p* = 0.010). No other associations regarding SctO2 and age were found in patients with negative HUT. The SBP levels among these tilt-negative patients in the supine position and after 3 and 10 min of HUT were associated with age (*p* < 0.001 for all three analyses) (Appendix A).

In VVS patients, SctO2 during the presyncopal phase was significantly lower in older patients (*p* = 0.009 for all group comparisons) (Table 2, Figure 3A). In contrast, mean SBP during the presyncopal phase did not differ by age group (*p* = 0.133) (Appendix A). Delta SctO2 presyncope was significantly greater in older VVS patients compared with younger patients (*p* = 0.003) (Table 2). Among OH patients, SctO2 was significantly lower in older subjects after 10 min of HUT, during the presyncopal phase (Figure 3B), and at minimum SctO2 (*p*< 0.05 for all group comparisons) (Table 3), whereas SBP during the presyncopal phase did not differ according to age (*p* = 0.810) (Appendix A). Moreover, the decrease in SctO2 from supine to presyncopal phase or minimum value differed according to age, being largest in the age group above 60 years (Table 3). Full data of the comparisons of SctO2 and SBP among groups are displayed in Appendix A.

## 4. Discussion

We found that older patients with vasovagal syncope and orthostatic hypotension had lower cerebral tissue oxygenation before head-up-tilt induced syncope compared with younger patients, independently of concurrent systolic blood pressure. Moreover, advanced age was associated with lower minimum cerebral tissue oxygenation among patients with orthostatic hypotension, even if not all patients experienced syncope during HUT. In contrast, there was no association between age and cerebral tissue saturation during orthostatic provocation among patients with a negative head-up-tilt test. Whereas the direct clinical implications of the results are not clear at this point, our current results provide important information on age differences to SctO2 among patients with VVS and OH. For example, future studies should investigate the possible connection between lower SctO2 and cognitive dysfunction, as well as amnesia for during syncope.

### 4.1. Age and Cerebral Tissue Oxygenation among Negative HUT Patients

To the best of our knowledge, this is the largest study on age-related differences in SctO2 in VVS and OH. Previous studies have explored cerebral circulation during orthostatic provocation in relation to aging in predominantly healthy subjects, but findings have been inconsistent [14,15,16,17]. We found no association between aging and SctO2 during orthostasis in patients with negative HUT, which is in accordance with previous reports of similar cerebral autoregulatory responses to orthostasis among older and younger healthy individuals [15,18]. A similar decline in CBF velocity in younger and older individuals during orthostatic provocation was found by Sorond et al. [16].

In contrast, studies have shown decreased frontal cortical oxygenation in healthy elderly compared with younger subjects [17], and a small decline in oxyhemoglobin during supine-to-standing, although no correlation with decreased CBF [19]. Group heterogeneity and differences in HUT or active standing test protocols may explain the inconsistency of previous results. Our findings demonstrate that aging is associated with lower SctO2 at rest, whereas SctO2 during orthostatic provocation remains intact, possibly implying a more preserved cerebral autoregulation due to better adaptation to blood pressure swings that is more often observed in these subjects [20]. It appears that cerebral autoregulation is rather well preserved in older age [21], although BP adaptation to, for instance, postural changes is impaired [20].

### 4.2. Lower Cerebral Tissue Oxygenation among Older VVS and OH Patients Prior to Syncope

Few studies have measured SctO2 in relation to age in VVS and OH patients. In this study, older VVS and OH patients had lower SctO2 during the presyncopal phase compared with younger patients. In addition, lower minimum SctO2 values were also associated with increasing age among OH patients, even if not all OH patients experienced syncope. We hypothesize that cerebral autoregulatory mechanisms may be impaired in older patients with VVS and OH. This may lead to adaption to lower SctO2 levels in standing, in turn resulting in a lower SctO2 threshold for when syncope occurs. One earlier study, based on the SYSTEMA cohort, including fewer patients (n = 54) with either VVS (n = 31), OH (n = 13) or patients with negative HUT (n = 10) found no effect of age on either SctO2 values at baseline or on the change in SctO2 from baseline to 1 min before reflex activation or end of tilt [13]. However, VVS, OH or negative HUT patients were not analyzed separately, which may explain the different results.

### 4.3. Older OH Patients Have Lower Cerebral Tissue Oxygenation during HUT

Interestingly, among OH patients, lower SctO2 was not only associated with older age during presyncopal phase but also when supine, after 3 and 10 min of HUT and at minimum SctO2, despite higher SBP at these times. This suggests that older OH patients have a greater degree of CA impairment compared with younger OH patients. It has previously been hypothesized that cerebral hypoperfusion in OH may play a role in the development of cognitive impairment. Previous studies have found that OH is associated with an overall prevalence of dementia or mild cognitive impairment [22]. Further, these studies have also concluded that OH in the elderly seems to be a risk factor for cerebral hypoxic damage and incident mild cognitive impairment, probably due to disturbed CA [22]. The lower SctO2 that is found among older OH patients might therefore be important when considering BP treatment and may emphasize the importance of avoiding “overtreating” BP among these vulnerable patients.

### 4.4. Clinical and Physiological Differences among Syncope Patients according to Age

The postural decreases in SctO2 that were present in older VVS and OH patients might indicate that regulation of SctO2 alters with increasing age. These findings suggest that CA fails to compensate completely for postural changes in older VVS and OH patients, which may predispose these patients to ischemic cerebral symptoms during standing. However, prodromal symptoms, both those due to autonomic activation and to retinal and cerebral hypoperfusion, are usually less obvious in older patients [23]. Moreover, VVS patients aged ≥60 years are less likely to report complete or near loss of consciousness and more likely to report unexplained falls than those aged <60 years. Furthermore, prodromal symptoms were less common in older patients [5]. Another study found that frontal cortical oxygenation declined remarkably in healthy elderly subjects after standing but without cerebral symptoms. Thus, even though older syncope and orthostatic intolerance patients are more susceptible to cerebral ischemia when upright, their clinical presentation seems inconsistent, possibly due to better hypoperfusion tolerance or impaired cognition [17], even though this is speculative for the time being. Accordingly, clinicians need to have a high degree of suspicion for syncope as a cause when evaluating older patients presenting collapse or unexplained falls, especially when the clinical history seems inconsistent or fragmented.

In addition to age related differences in the clinical presentation of syncope and OH, BP tends to fall more slowly, and bradycardia is usually less pronounced in older VVS patients [24]. A possible explanation as to why older patients in our study seem to tolerate deoxygenation during orthostatic provocation better than younger patients may be that abrupt cessation of cerebral circulation leads to loss of cerebral function in young adults after about 7 s [23]. This period, cerebral anoxia reserve time, is shorter in young individuals and longer in the elderly [23]. However, even though older patients seem to tolerate deoxygenation better in terms of manifest syncope, the study of cognitive function in relation to deoxygenation during the presyncopal phase would be an interesting subject for further research.

### 4.5. Limitations

This study has some important limitations. First, even though the negative HUTs consisted of patients with a normal response to orthostatic provocation, they were all patients with a history of syncope and/or orthostatic intolerance.

Second, polypharmacy and multimorbidity increases with age [6]. In this study, we did not adjust for such factors in the statistical analysis. Future studies should investigate the role of polypharmacy and multimorbidity on SctO2 and their role in developing syncope and orthostatic intolerance.

Third, OH patients were older than negative-HUT patients, making comparisons difficult.

Fourth, NIRS harbors some technical limitations: Cerebral tissue saturation levels were measured in the regions where the probes were located, with a penetration depth of 2.5 cm. Thus, SctO2 in other or deeper parts of the brain was lacking. This limitation of NIRS has been more extensively discussed elsewhere [7]. Furthermore, according to previous research [25], changes in myelination of the cerebral cortex with age produces a greater dispersion of infrared light passing through the brain, which may explain the influence of age on baseline SctO2. However, in this study, the decrease in SctO2 from baseline to the presyncopal phase was greater among older VVS and OH patients compared with younger patients, which suggested that the observed difference was not only due to age related changes in myelination. A further limitation of NIRS is the risk of measuring the saturation of overlying tissues or deeper regions of the brain [7].

Fifth, previous studies have found that haemoglobin levels are higher among smokers [26], which could possibly also have an impact on Scto2 levels. Previous studies have found that lower haemoglobin levels are associated with lower SctO2 levels [27,28]. In the current study, we did not have data on haemoglobin levels, which may be a. However, we did not find a significant difference in supine SctO2 levels among smokers vs. non-smokers, which may indicate that smoking habits did not have a major impact on our results.

## 5. Conclusions

Older patients with vasovagal syncope and orthostatic hypotension have lower cerebral tissue oxygenation in the presyncopal phase compared with younger patients, independently of concurrent systolic blood pressure. In contrast, there was no association between age and cerebral tissue oxygenation during orthostatic provocation among patients with a negative response to head-up tilt. These results suggest that during imminent reflex syncope, cerebral tissue oxygenation decreases more with advancing age. Future studies should further investigate the implications of these findings.

## Figures and Tables

**Figure 1 jcm-11-04302-f001:**
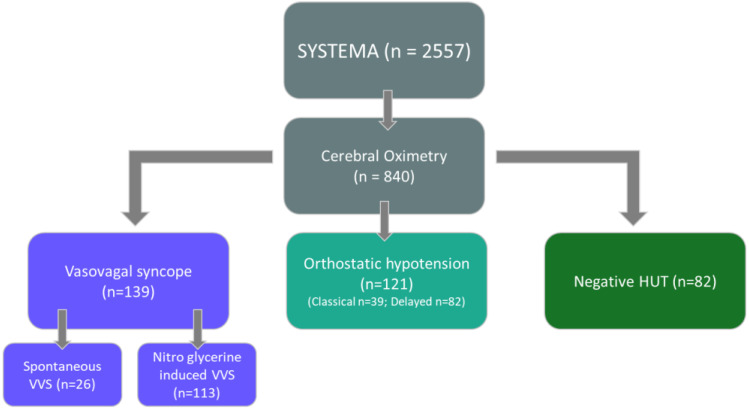
Study population flow chart. Abbreviations: HUT = head-up tilt test; SYSTEMA = syncope study of unselected population in Malmö; VVS = vasovagal syncope.

**Figure 2 jcm-11-04302-f002:**
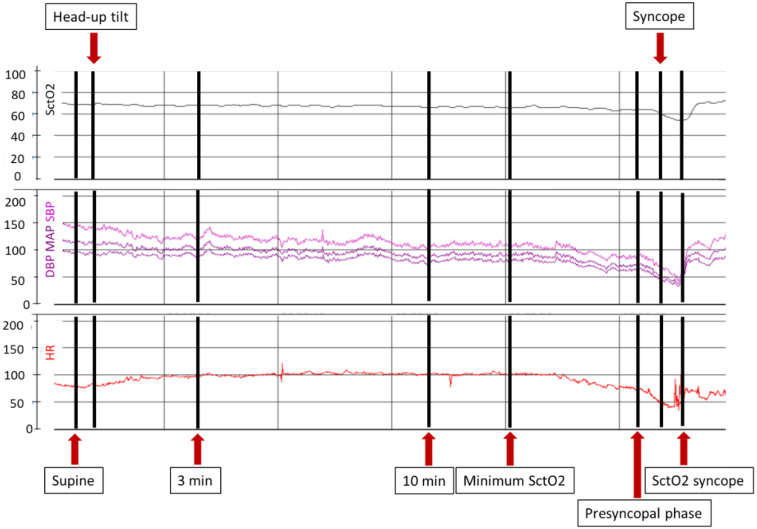
Example of a patient with vasovagal syncope during head-up tilt test. SctO2 values were collected in supine position, after 3 and 10 min of tilt, during lowest measured value before reflex activation (minimum SctO2); 30 s before syncope (presyncopal phase); and lowest measured value during syncope (SctO2 syncope). Abbreviations: DBP = diastolic blood pressure; HR = heart rate; MAP = mean arterial pressure; SBP = systolic blood pressure; SctO2 = cerebral tissue oxygenation.

**Figure 3 jcm-11-04302-f003:**
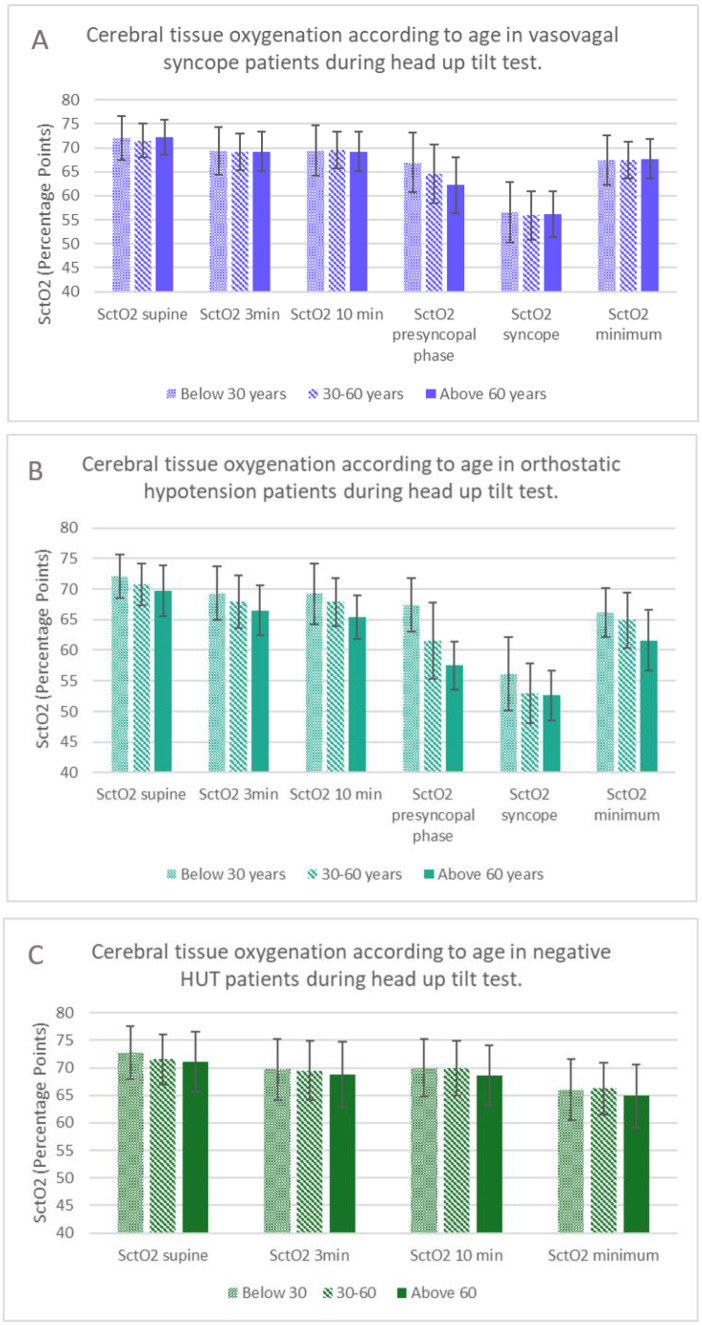
Cerebral tissue oxygenation according to age in VVS, OH and negative head-up tilt tests. (**A**). SctO2 during HUT in VVS. (**B**). SctO2 during HUT in OH. (**C**). SctO2 during HUT in patients with negative HUT. Abbreviations: HUT = head-up tilt test; OH = orthostatic hypotension; SctO2 = cerebral tissue oxygenation; VVS = vasovagal syncope.

**Table 1 jcm-11-04302-t001:** Baseline characteristics and hemodynamic parameters in VVS, OH and negative head-up tilt tests.

	VVS (n = 139)	OH (n = 121)	Negative HUT (n = 82)	*p*-Value
Baseline Characteristics				
Age (years)	45.1 ± 17.1	61.4 ± 19.2	44.5 ± 18.2	<0.001
Women (%)	59.7	49.6	61.0	0.165
Hypertension (%)	20.7	36.4	12.9	<0.001
Ischemic heart disease (%)	4.3	6.6	6.1	0.701
Stroke (%)	1.4	11.6	2.4	<0.001
Heart failure (%)	1.4 *	4.4	6.1	0.090
Atrial fibrillation (%)	1.4	12.4	2.4	<0.001
Diabetes mellitus (%)	6.5	12.4	9.8	0.260
Current smoker (%)	10.8	14.9	27.2 *	0.006
Resting HR (bpm)	66.3 ± 11	70.5 ± 11.1	70.6 ± 11.5	0.004
Resting SBP (mmHg)	131.3 ± 16.4	142.0 ± 22.6	134.0 ± 17.7	<0.001
SctO2				
Supine (pp)	72.0 ± 3.9	70.2 ± 4.0	71.5 ± 4.8 †	0.003
3 min (pp)	69.2 ± 4.2 †	67.2 ± 4.3 †	69.4 ± 5.5	<0.001
10 min (pp)	69.5 ± 4.2 ^§^	66.4 ± 4.0 ^#^	69.6 ± 5.1	<0.001
Syncope (pp)	56.1 ± 5.3	53.3 ± 4.7 **		
Minimum (pp)	67.5 ± 4.2	63.0 ± 5.1	65.9 ± 5.2 ^‡^	<0.001
Presyncopal phase (pp)	64.6 ± 6.2	60.3 ± 5.8 **		<0.001
Delta presyncope	7.3 ± 5.2	9.5 ± 4.2 **		
HR				
3 min (bpm)	78.2 ± 11.5 †	80.9 ± 13.8 †	79.2 ± 13.7	0.250
10 min (bpm)	79.1 ± 11.6 ^§^	83.2 ± 14.5 ^||^	79.7 ± 13.7	0.050
Max (bpm)	83.6 ± 13.9 †	86.8 ± 16.8*	83.8 ± 14.8	0.205
Presyncopal phase (bpm)	85.8 ± 20.2	84.0 ± 23.5**		0.612
SBP				
3 min (mmHg)	133.6 ± 18.2 ‡	127.3 ± 22.3 †	138.2 ± 19.2	0.001
10 min (mmHg)	130.9 ± 17.7 ^§^	117.3 ± 22.4 ^||^	134.9 ± 19.2	<0.001
Minimum (mmHg)	121.2 ± 17.0 †	98.4 ± 20.4 *	127.2 ± 17.1	<0.001
Presyncopal phase (mmHg)	79.7 ± 20.6	72.5 ± 13.7 **		0.035

Continuous variables are expressed as mean ± standard deviation unless other specified. Dichotomous data are expressed as percentages of total within each group. *p*-values denote overall differences from one-way ANOVA for continuous data and Pearson’s chi^2^-test for dichotomous data. Delta presyncope was defined as the difference in supine value to 30 s before syncope. SctO2 syncope was defined as lowest measured value during syncope. SctO2 minimum, SBP minimum and HR max are defined as lowest or highest value before reflex activation. * n = 1, † n = 2, ‡ n = 3, § n = 11, ^||^ n = 14, # n = 15, ** n = 78 missing. HR = heart rate; HUT = head-up tilt; OH = orthostatic hypotension; SBP = systolic blood pressure; SctO2 = cerebral tissue oxygenation; VVS = vasovagal syncope; pp = percentage points; acute coronary syndrome = previous acute myocardial infarction, unstable angina, percutaneous coronary intervention and/or coronary artery bypass graft surgery.

**Table 2 jcm-11-04302-t002:** Cerebral tissue oxygenation according to age in VVS patients during head-up tilt test.

	Below 30 Years (n = 36)	30–60 Years (n = 73)	Above 60 Years (n = 30)	*p*-Value
SctO2 supine	72.8 ± 4.6	71.5 ± 3.5	72.2 ± 3.6	0.256
SctO2 3 min	69.4 ± 5.0	69.1 ± 3.8 *	69.2 ± 4.1	0.953
SctO2 10 min	69.4 ± 5.3 ‡	69.6 ± 3.8 †	69.2 ± 4.1	0.907
SctO2 presyncopal phase	66.9 ± 6.2	64.5 ± 6.1	62.2 ± 5.8	0.009 ^β^
SctO2 syncope	56.5 ± 6.3	55.9 ± 5.0	56.1 ± 4.8	0.865
SctO2 minimum	67.4 ± 5.1	67.5 ± 3.8	67.7 ± 4.1	0.975
SctO2 delta presyncope	5.9 ± 4.4	6.9 ± 5.2	10.0 ± 5.1	0.003 ^β^
SctO2 delta syncope	16.3 ± 6.2	15.6 ± 4.4	16.1 ± 3.9	0.748 ^α^
SctO2 delta minimum	5.3 ± 3.4	4.0 ± 2.0	4.5 ± 2.7	0.091 ^α^

Continuous variables are expressed as mean ± standard deviation. *p*-values are from one-way ANOVA, unless otherwise indicated, for differences in mean SctO2 in patients <30 years, 30–60 years and >60 years of age. Delta presyncope was defined as the difference in supine value to 30 s before syncope. Delta syncope was defined as the difference in supine value and lowest measured SctO2 during syncope. Delta minimum was defined as the difference in supine value and minimum value before reflex activation. * n = 2, † n = 5, ‡ n = 6 missing. SctO2 = cerebral tissue oxygenation; VVS = vasovagal syncope. α = *p*-values from Welch test. β: Tukey’s post hoc; SctO2 presyncopal phase <30 vs. >60 years: *p* < 0.05; SctO2 delta presyncope <30 vs. >60 and 30–60 vs. >60: *p* < 0.05.

**Table 3 jcm-11-04302-t003:** Cerebral tissue oxygenation according to age in OH patients during head-up tilt test.

	Below 30 Years (n = 13)	30–60 Years (n = 32)	Above 60 Years (n = 76)	*p*-Value
SctO2 supine	72.1 ± 3.6	70.8 ± 3.4	69.7 ± 4.2	0.090
SctO2 3 min	69.3 ± 4.4 *	68.0 ± 4.3	66.5 ± 4.1 *	0.050
SctO2 10 min	69.2 ± 5.0 †	67.9 ± 3.9 †	65.4 ± 3.6 ^§^	0.001 ^β^
SctO2 presyncopal phase	67.4 ± 4.4 ‡	61.6 ± 6.2 ^||^	57.5 ± 3.9 ^#^	<0.001 ^β^
SctO2 syncope	56.1 ± 6.0 ‡	52.9 ± 4.9 ^||^	52.6 ± 4.1 ^#^	0.206
SctO2 minimum	66.2 ± 4.0	64.9 ± 4.5	61.6 ± 5.0	<0.001 ^β^
SctO2 delta presyncope	5.4 ± 4.7 ‡	8.6 ± 3.5 ^||^	11.1 ± 3.5 ^#^	0.003 ^β^
SctO2 delta syncope	16.7 ± 5.5 ‡	17.3 ± 2.8 ^||^	16.0 ± 4.2 ^#^	0.701
SctO2 delta minimum	5.9 ± 3.0	5.9 ± 3.2	8.1 ± 4.4	0.018 ^β^

Continuous variables are expressed as mean ± standard deviation. *p*-values are from one-way ANOVA for differences in mean SctO2 in patients <30 years, 30–60 years and >60 years of age. Delta presyncope was defined as the difference in supine value to 30 s before syncope. Delta syncope was defined as the difference in supine value and lowest measured SctO2 during syncope. Delta minimum was defined as the difference in supine value and minimum value before reflex activation. * n = 1, † n = 3, ‡ n = 6, § n = 9, || n = 20, # n = 52 missing. OH = orthostatic hypotension; SctO2 = cerebral tissue oxygenation. β: Tukey’s post hoc; SctO2 10 min and SctO2 minimum, <30 vs. >60 years and 30–60 vs. >60 years: *p* < 0.05; SctO2 presyncopal phase *p* < 0.05 between all three age groups; SctO2 delta presyncope <30 vs. >60 years, *p* < 0.05; SctO2 delta minimum, 30–60 vs. >60 years: *p* < 0.05.

## Data Availability

The data that support the findings of this study are available on request from the corresponding author (I.K.).

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
