# Peer review of "The Influence of Age on Cerebral Tissue Oxygenation in Vasovagal Syncope and Orthostatic Hypotension"

_jcm, 2022, doi:10.3390/jcm11154302_

Round 1

Reviewer 1 Report

Dear Sir/Madam,

I had the opportunity to act as a reviewer on the recent submission by Kharraziha et al. to the Journal of Clinical Medicine.

The authors present an interesting original article investigating the age-related differences in cerebral tissue oxygenation in reflex and orthostatic syncope. They have included 139 patients with reflex syncope, 121 patients with orthostatic syncope and 82 patients with negative head-up tilt test. The manuscript is well written and the results are interesting.

However, some major issues need to be addressed:

  1. The syncope guideline of the European Society of Cardiology since 2009 recommends (class I) the carotid sinus massage as part of the non-invasive assessment of syncope. Could you provide the results of the carotid sinus massage in all patients?
  2. The table with baseline characteristics includes no details about the comorbidities of the patients. This is of utter importance, especially in older patients with advanced carotid and/or diabetic disease. This important effects on the cerebral autoregulation. Please add in the table with baseline characteristics the comorbidities.
  3. How do the authors interpret the findings? What could be the pathophysiological explanation for the results? Please comment in the Discussion section.
  4. What are the clinical implications of the findings, how can the findings of the study change or improve the daily practice? Please comment.

Minor issues:

  1. The abbreviation “HUT” is used in the abstract without defining it first. Please explain the abbreviation at its first use in the abstract.
  2. The Table 1 from the supplementary material is difficult to understand. It would help explaining the units of measurement.

Best regards

Reviewer 2 Report

I read with pleasure this original article written by Dr. Kharraziha and colleagues. The study explored the differences of central SctO2 across age strata in vasovagal and orthostatic syncope. The Authors used a large dataset which included data gathered in a single Swedish centre. Age cutoff were derived from previous observations and seem thus clinically impactful.

They found that older patients have lower SctO2 compared to younger individuals, both in case of VVS or OH.

The article is well written and interesting, statistical analysis is simple but effective. Despite the single centre nature of the study being a limitation, the sample is large and derives from a solid dataset.  I have only some minor comments:

- If possible, I would suggest adding a figure with the longitudinal variations of the SctO2, substantially summarizing most findings reported in table 2B. I think this may be of impact and would allow a more easy understanding of some interesting study findings.

- Abstract: the abbreviation “HUT” is never explained.

Round 2

Reviewer 1 Report

Dear Sir/Madam,

Thank you for reviewing the manuscript and addressing the mentioned issues. These were adequately answered. Therefore, the manuscript seems suitable for publishing in the present form.

Best regards